# Assessment of Protein Content and Phosphorylation Level in *Synechocystis* sp. PCC 6803 under Various Growth Conditions Using Quantitative Phosphoproteomic Analysis

**DOI:** 10.3390/molecules25163582

**Published:** 2020-08-06

**Authors:** Masakazu Toyoshima, Yuma Tokumaru, Fumio Matsuda, Hiroshi Shimizu

**Affiliations:** Department of Bioinformatics Engineering, Graduate School of Information Science and Technology, Osaka University, 1-5 Yamadaoka, Suita, Osaka 565-0871, Japan; toyoshima104@ist.osaka-u.ac.jp (M.T.); yuma_tokumaru@bio.eng.osaka-u.ac.jp (Y.T.); fmatsuda@ist.osaka-u.ac.jp (F.M.)

**Keywords:** phosphoproteome, phycobilisome, *Synechocystis* sp. PCC 6803, targeted proteomics

## Abstract

The photosynthetic apparatus and metabolic enzymes of cyanobacteria are subject to various controls, such as transcriptional regulation and post-translational modifications, to ensure that the entire cellular system functions optimally. In particular, phosphorylation plays key roles in many cellular controls such as enzyme activity, signal transduction, and photosynthetic apparatus restructuring. Therefore, elucidating the governing functions of phosphorylation is crucial to understanding the regulatory mechanisms underlying metabolism and photosynthesis. In this study, we determined protein content and phosphorylation levels to reveal the regulation of intracellular metabolism and photosynthesis in *Synechocystis* sp. PCC 6803; for this, we obtained quantitative data of proteins and their phosphorylated forms involved in photosynthesis and metabolism under various growth conditions (photoautotrophic, mixotrophic, heterotrophic, dark, and nitrogen-deprived conditions) using targeted proteomic and phosphoproteomic analyses with nano-liquid chromatography-triple quadrupole mass spectrometry. The results indicated that in addition to the regulation of protein expression, the regulation of phosphorylation levels of cyanobacterial photosynthetic apparatus and metabolic enzymes was pivotal for adapting to changing environmental conditions. Furthermore, reduced protein levels of CpcC and altered phosphorylation levels of CpcB, ApcA, OCP, and PsbV contributed to the cellular response of the photosynthesis apparatus to nitrogen deficiency.

## 1. Introduction

Cyanobacteria convert light energy into ATP and NADPH through photosynthesis and use it in metabolic reactions to fix CO_2_. Various intracellular reactions, such as light harvesting, photosynthetic electron transfer, CO_2_ fixation, and metabolic reactions, are stringently regulated and optimized with regard to the surrounding environment. Photosystems and Rubisco are the most abundant proteins in cyanobacteria, and their gene expression levels fluctuate greatly in response to nutrient and light conditions. However, mRNA and protein levels of only 10–15% genes in the cyanobacterium *Synechocystis* sp. PCC 6803 are correlated; specifically, the mRNA and protein levels of photosynthesis-related genes are rarely correlated, and their post-transcriptional regulation is complex [1]. Furthermore, protein accumulation and activity are closely regulated by post-translational modifications (PTMs) [2].

Post-translational modifications (PTMs), such as phosphorylation, acetylation, and methylation, are an important process that regulates protein activity [2]. Among these, phosphorylation is a highly conserved PTM in several organisms from prokaryotes to eukaryotes, and it is closely associated with intracellular metabolism and signaling [3]. Phosphorylation usually occurs on serine (Ser), threonine (Thr), and tyrosine (Tyr) residues of eukaryotic proteins. In addition, in prokaryotic proteins, phosphorylation also occurs on histidine, arginine, and lysine residues [4]. Phosphorylation leads to the formation of a phosphate ester bond in the hydroxyl groups of amino acids, such as Ser, Thr, and Tyr, in proteins, and alters the protein structure by imparting a negative charge following the addition of a phosphate group. Citrate dehydrogenase in the tricarboxylic acid (TCA) cycle is a well-known enzyme that is inactivated by phosphorylation and activated by dephosphorylation [5]. In cyanobacteria, a signal protein called PII is phosphorylated under nitrogen-limited conditions, and it then controls metabolism by facilitating the expression of genes encoding enzymes involved in nitrogen assimilation, the pentose phosphate pathway, and glycolysis [6,7]. Regarding the photosynthetic apparatus, turn-over of the D1 protein in *Arabidopsis* and remodeling of the photosystem II (PSII)–LHCII supercomplex in *Chlamydomonas* have also been reported to be regulated by phosphorylation [8,9]. Thus, phosphorylation is a key PTM involved in many cellular controls, such as enzyme activity, signal transduction, and photosynthetic apparatus restructuring, and elucidating the governing roles of phosphorylation is imperative to understand the regulatory mechanisms underlying metabolism and photosynthesis. The genome of *Synechocystis* sp. PCC 6803 encodes 11 Ser/Thr kinase, 1 Tyr kinase, and 7 phosphatases [10], and approximately 5% of the proteins in the cell are phosphorylated [11]. A comprehensive shotgun phosphoproteome of *Synechocystis* has been established [11,12,13,14,15]. However, quantitative analysis of this cyanobacterium is limited by the low levels of phosphoproteins present in the cells and their susceptibility to ionization suppression during liquid chromatography-triple quadrupole mass spectrometry (LC-MS/MS) [16].

In this study, target proteomics was used to quantitatively analyze the phosphorylation levels of Ser, Thr, and Tyr residues of specific proteins. To elucidate the regulation of intracellular metabolism and photosynthesis by protein expression and phosphorylation levels, we obtained data of the expression of proteins and their phosphorylation levels during photosynthesis and metabolism under various growth conditions (photoautotrophic, mixotrophic, heterotrophic, dark, and nitrogen-deprived conditions). Based on the obtained data, we demonstrated changes in protein expression and phosphorylation levels of the photosynthetic apparatus and metabolic enzymes under these growth conditions. Furthermore, we evaluated the association of regulation of protein expression and phosphorylation with adaptability to different environments.

## 2. Results

### 2.1. *Synechocystis sp.* PCC 6803 Culture under Various Growth Conditions

Figure 1b shows the growth curves of *Synechocystis* sp. PCC 6803 under five nutrient conditions: (1) photoautotrophic (Auto; photosynthesis only), (2) mixotrophic (Mixo photosynthesis and glucose utilization), (3) heterotrophic (Hetero; glucose utilization only), (4) dark (Dark), and (5) nitrogen deprived (–N). Logarithmic growth was the fastest under the Mixo condition, followed by that under Hetero and Auto conditions. The specific growth rates under Auto and Hetero conditions were about half and two-thirds of that under the Mixo condition, respectively. Under −N, there was no significant reduction in growth rate up to 48 h after nitrogen deprivation, but after 72 h, growth almost stopped and reached a plateau.

The absorption spectra of cell suspensions at each logarithmic growth phase were examined (Figure 1c). Compared to the Auto condition, the Dark and Mixo conditions showed no significant changes. Meanwhile, consistent with previous reports, under N24 and N48, the absorption peak of phycocyanin (630 nm), a pigment contained in phycobilisome, significantly decreased, and the color of the cell culture turned yellow (Figure 1c,d). Absorbance at this wavelength increased under the Hetero condition. In fact, under the Hetero condition, the culture medium turned dark green, which is consistent with observations in a previous study [17].

### 2.2. Acquisition of Targeted Proteomic and Phosphoproteomic Data

For targeted proteomic analysis, aliquots of the culture suspension were collected at the logarithmic growth phase (OD_730_ ~ 1.0). Proteins were extracted and digested with trypsin, and the tryptic peptide samples were subjected to nano-liquid chromatography-triple quadrupole mass spectrometry (LC-MS/MS) with multiple reaction monitoring (MRM) assays. Using the MRM assays established in our previous study [18], 197 central metabolic and photosynthesis-related proteins were analyzed. In the present proteomic analysis, 128 proteins (58.0%) were successfully quantified (Appendix A). The expression levels were measured as a quantitative value relative to the value under the Auto condition (Figure 2a). Proteomic data obtained under the Auto, Mixo, and Hetero conditions were compared with gene expression data reported in previous studies [19] (Appendix A). Correlation coefficients of the proteomic data of relative change in gene expression under Mixo/Auto and Hetero/Auto conditions were *r* = 0.09 and 0.12, respectively. Consistent with previous reports, there was no clear correlation between gene expression and protein level. In addition, the correlation between our proteomic data and flux values of metabolic reactions reported in a previous study [20] was also examined (Appendix A) but no clear correlation was observed for many proteins. This result suggests that many metabolic reactions are controlled by PTMs, such as phosphorylation, rather than by the levels of metabolic enzymes. Furthermore, in addition to metabolic enzymes, many photosynthesis-related proteins have been reported to be phosphorylated [12,14].

Therefore, to quantitatively analyze the regulatory functions of phosphorylation under various growth conditions, we attempted to analyze the phosphoproteomic data using nanoLC–MS/MS. For 101 proteins whose phosphorylation sites have been identified in previous studies [12,14], we attempted to develop MRM assays and successfully determined the relative levels of 31 phosphopeptides in 24 proteins (Appendix A). The relative quantitative values of phosphopeptides were converted to phosphorylation rates by dividing with values of proteins. The phosphorylation rate was presented as a quantitative value relative to the Auto condition (Figure 2b). Twelve of the successfully quantified proteins were photosynthesis-related, such as antenna proteins.

### 2.3. Comparison of Protein Expression and Phosphorylation Profiles

The protein expression profile showed that under all growth conditions, the variation in phosphorylation level was greater than that in protein level, and the phosphorylation state changed widely under each condition (Figure 3a). Therefore, principal component analysis (PCA) was performed on the proteomic and phosphoproteomic data (Figure 3b,c). The score plot of proteomic data showed that Dark and Mixo condition data were clustered near Auto condition data, indicating little differences in expression profiles among these data. The expression profiles of N24 and N48 changed significantly in the direction of the first principal component. The expression profile under the Hetero condition also changed in the direction of the first principal component (Figure 3b). In fact, N24 and N48 showed similar overall protein expression profiles (Figure 2a). In contrast, the score plots of phosphoproteomic data showed that the Dark and Mixo condition data were cultured away from the Auto condition data (Figure 3c). These results suggest that in the Dark and Mixo condition data, although the variation at the proteome level was small, variation at the phosphoproteome level was significant.

Under the Dark condition, the level of the allophycocyanin core linker protein ApcC increased (by 2.7 fold) and that of the photosynthesis-related protein PsbP decreased (by 0.4 fold). Conversely, the phosphorylation profiles changed significantly. Phosphorylation levels of the Rubisco large subunit RbcL (Slr0009), Rubisco small subunit RbcS (Slr0012), glycogen-metabolizing enzyme Pgm (Sll0726), and glucose metabolism transcriptional regulator anti-sigB (Slr1856 and Slr1859), among others, were increased (Figure 2b). These results suggest that the photosystem and metabolic enzymes are regulated through phosphorylation under Dark conditions.

Under the Hetero condition, a significant decrease in the expression of RbcL and RbcS, which catalyze carbon fixation, and of CcmK1 and CcmK2, which are involved in CO_2_ concentrating mechanism, was observed as a change at the proteome level. This is consistent with a reduction in the flux of CO_2_ fixation by Rubisco under Hetero conditions in a previous study using ^13^C-MFA analysis [20]. In addition, the expression of PSII and ATPase increased (Figure 2a). Under the Hetero condition in our study, the addition of 3-(3,4-dichlorophenyl)-1,1-dimethylurea (DCMU) to the medium for blocking electron transfer from PSII to plastoquinone led to the inhibition of photosynthesis, and the cells used only glucose added to the medium even in the presence of light. This led to an over-reduced state of PSII, which likely increased the expression of PSII proteins for preventing injuries to PSII. Moreover, under this condition, the reduction power generated by glucose as a substrate reduced the plastoquinone pool through glycolysis and the TCA cycle, and ATP was synthesized via the respiratory chain electron transport.

Furthermore, state transition (when the absorbed light energy is distributed to PSI) occurred following light irradiation in the presence of DCMU; this resulted in the cyclic electron transport being driven, which ultimately activated ATP synthesis. These reactions may have increased ATPase expression. At the phosphoproteome level, the phosphorylation level of RbcL and RbcS was elevated but that of anti-sigB was reduced under Dark condition (Figure 2b).

Under −N (N24 and N48) conditions, significant changes compared to those under Auto condition were observed at both proteome and phosphoproteome levels. The protein levels of glycogen-degrading enzymes, such as GlgP and GlgX, and enzymes involved in the pentose phosphate pathway and glycolysis, such as Zwf, Gnd, and Gap1, were increased. In addition, the expression of GlnN, a nitrogen assimilating enzyme, was increased by over 80 fold. Moreover, phosphorylation levels of the PII protein (Ssl0707) were increased. These results are consistent with previous reports on the regulation of PII protein by glycogen degradation induced via the σ factor SigE and C/N balance in *Synechocystis* sp. PCC 6803 under nitrogen-depleted conditions [7,12,21]. Among photosynthesis-related proteins, the levels of PsbV and PsaD were increased at both proteome and phosphoproteome levels. In addition, the phosphorylation level of orange carotenoid-binding protein (OCP, Slr1963), a water-soluble protein containing carotenoids that is involved in non-photo quenching (NPQ) and that dissipates excess light energy as heat [22], was increased under nitrogen deficiency (Figure 2b).

Under −N (N24 and N48) conditions, the absorption peak of phycobilisome in the cell suspension was reduced, leading to cell chlorosis (Figure 1c,d). Apparently, phycobilisomes are mostly lost under nitrogen deficiency. Interestingly, among the antenna proteins, only the levels of CpcC and CpcC2, which are linker proteins between phycocyanin (PCs), were significantly reduced. The abundance of CpcC and CpcC2 under N48 was drastically reduced to 8% and 36%, respectively. In contrast, the abundance of ApcC was increased by 5.7 fold. At the phosphoproteome level, the phosphorylation levels of CpcB, the β-subunit of PC (β-PC), increased (Figure 4a). In ApcA, which is the α-subunit of allophycocyanin (α-APC), the levels the phosphopeptide YLSPGELDR were increased but those of SLGTPIEAVAQSVR and IAETLTGSRETIVK were decreased (Figure 2b). These results indicate that under nitrogen deficiency, the protein levels of individual subunits of the phycobilisome complex were not reduced, although the structure of the phycobilisome complex may be dissociated because of reduced levels of CpcC and altered phosphorylation states of CpcB and ApcA.

### 2.4. Changes in Photosynthetic Parameters under Nitrogen Deficiency

Proteomic analysis under −N conditions showed partial degradation of the phycobilisome complex and regulation of phosphorylation levels. Therefore, different photosynthetic parameters under −N (N24 and N48) conditions were evaluated using pulse-amplitude modulation (PAM) fluorescence measurements (Table 1). The *F*_v_/*F*_m_, the maximum quantum yield of PSII, was 0.515 ± 0.012 under Auto condition, which was almost the same as that under −N conditions. However, a previous study has suggested that the “true” maximum quantum yield of PSII was reduced to approximately half under nitrogen deficiency [23]. Meanwhile, qN, an index of NPQ of light energy dissipated as heat and state transition of phycobilisome, decreased in a time-dependent manner under −N conditions. Similarly, qP, an index of photochemical quenching, decreased under −N conditions, indicating that Q_A_ is reductive. Accordingly, ΦII, the effective quantum yield of electron transfer of PSII, decreased.

## 3. Discussion

Cyanobacteria have a photosynthetic apparatus and metabolic enzymes for carbon fixation, and their biosynthesis and metabolism require numerous amino acids and resources. The roles of PTMs, such as phosphorylation, are also pivotal because the environment can dramatically change in a short period. In this study, to elucidate their regulation and roles, the levels of proteins involved in photosynthesis and metabolism as well as their phosphorylation states under various growth conditions were analyzed using targeted proteomics.

Our results showed that the expression levels of only certain proteins changed in response to growth conditions. For example, under the Hetero condition, although photosynthesis was terminated, the levels of only RbcL, RbcS, and CO_2_-concentrating mechanism proteins among the metabolic enzymes were drastically reduced. However, the levels of other enzymes in the Calvin cycle did not change significantly (Figure 2a). As RbcL and RbcS are the most abundant proteins in the cell, reducing the excess RbcL and RbcS levels under the Hetero condition, where photosynthesis was inhibited, would greatly reduce the amount of amino acids required for protein biosynthesis.

Under nitrogen deficiency, cells lose phycobilisomes, a huge protein complex, and redistribute nitrogen to other highly important metabolites [24]. In fact, the results of absorption spectra, presented in Figure 1c, showed that the absorption peak of phycobilisome at 630 nm was drastically reduced, indicating phycobilisome degradation. However, not all subunits of the phycobilisome complex were uniformly degraded; levels of only CpcC, a rod-linker protein of the phycobilisome complex, were drastically reduced. This result indicates that the phycobilisome complex possibly dissociates under nitrogen deficiency, but the individual subunits remain intact, allowing resumption of photosynthesis as soon as the growth environment improves.

In contrast, phosphorylation levels varied greatly than protein levels under different growth conditions (Figure 3a, Appendix A). Regulation through PTMs, such as phosphorylation, may play an important role. In particular, little changes at the proteome level were observed under Dark condition (Figure 2a). A previous study has reported that mRNA expression levels of photosynthesis-related genes decreased between 1 and 9 h after transition from light to dark [25]. However, the effects of protein synthesis and degradation were still considered to be negligible because the change in protein levels was small under Dark condition after 6 h (Figure 2a, Figure 3b). In contrast, the phosphorylation levels of proteins involved in metabolism, such as Pgm, CcmM, RbcL, RbcS, anti-sigB1, and anti-sigB2, were evidently altered. These results suggest that the photosynthetic apparatus and metabolic enzymes are maintained at the proteome level in dark conditions and are regulated via phosphorylation such that cellular activity can be resumed immediately under light conditions.

Furthermore, the phosphorylation levels of CpcB, ApcA (Y17 or S19), OCP, and PsbV were elevated under nitrogen deficiency (Figure 4). CpcB and ApcA construct the rod and core of the phycobilisome complex, respectively. Under nitrogen deficiency, the phycobilisome complex is degraded by Clp proteases and a small protein called NblA [26,27,28,29,30]. During this process, NblA binds to the *N*-terminal side of CpcB and ApcA and then dissociates the phycobilisome complex [27,30]. The phosphorylation sites of CpcB and ApcA detected in this study are located on the *N*-terminal side, and CpcB and ApcA phosphorylation may be involved in their binding with NblA. In addition, the phosphorylation levels of S118 or T121 in ApcA were reduced. This amino acid is located in the trimeric interaction domain region of ApcA [31]. Therefore, phosphorylation of this amino acid is involved in ApcA trimerization, and its dephosphorylation under nitrogen deficiency may be involved in dissociation of the ApcA trimer. OCP, a water-soluble carotenoid-binding protein, has been suggested to be involved in NPQ, which dissipates excess light as heat [22]. PsbV is a small subunit of PSII and contributes to the stabilization of the manganese cluster of PSII. Moreover, in *Synechocystis* sp. PCC 6803, PsbV suppresses PSII activity under nitrogen deficiency [1,32,33,34]. The *psbV*-disrupted strains maintain PSII activity even under nitrogen deficiency, while cell proliferation is reduced, indicating that the suppression of PSII activity by PsbV plays an important role in growth under nitrogen deficiency [1]. Our results, together with the previous reports, suggest that PsbV phosphorylation can suppress PSII activity under nitrogen deficiency.

Although analysis of the deficient strains and other factors are required to obtain direct evidence, based on the results of our proteomic and phosphoproteomic analyses as well as PAM measurements, a mechanism for adaptation to nitrogen deficiency was proposed, as illustrated in Figure 5. First, degradation of phycobilisome via phosphorylation and suppression of PSII photosynthetic activity by PsbV phosphorylation may occur. Although our results showed no difference in *F*_v_/*F*_m_ values between Auto and −N conditions, Ogawa et al. [23] have reported that *F*_v_/*F*_m_ under nitrogen deficiency is about half of the value under Auto condition. Moreover, Ogawa et al. [23] have reported that the estimation of photosynthesis by PAM measurement of cyanobacteria is always problematic owing to the interference from respiratory electron transfer and phycocyanin fluorescence; however, subtracting basal phycobilisome and PSI fluorescence allows to accurately estimate the maximum quantum yield (ΦII) of PSII, and difference between the “apparent” and “true” maximum quantum yield of PSII becomes larger under high phycocyanin conditions. Our PAM measurement system could not measure the “true” maximum quantum yield of PSII and measured the “apparent” maximum quantum yield. However, the results reported by Ogawa et al. [23] support our findings at the proteome and phosphoproteome levels. In other words, the uptake of light may be reduced at the entrance, such as phycobilisome, under nitrogen deficiency. ΦII and qP under nitrogen deficiency were lower than those under Auto condition, indicating that the rates of photosynthetic electron transfer and CO_2_ fixation were suppressed. In nitrogen-deficient cells, carbonate fixation was reduced owing to the blockade of metabolism caused by the C/N balance. In other words, the amount of energy required for the output seemed to have decreased.

Under −N (N24 and N48) conditions, qN and NPQ were reduced (Table 1). This result suggests that the rate of dissipation of light energy as heat was reduced and the efficiency of input was increased. Under −N (N24 and N48) conditions, the PQ pool is drastically reduced compared with that under Auto condition, even at the same light intensity because of delayed metabolism. This notion is supported by decreased qP under nitrogen deficiency. Therefore, under nitrogen deficiency, cells are stressed, similar to that under high light conditions, and become acclimatized to higher light intensities. It seems that NPQ under nitrogen deficiency was lower than that under Auto condition, because PAM measurements were performed at an actinic light intensity of 80 μm^−^^2^·s^−^^1^ and cultivation was performed under a light intensity of 50 μm^−^^2^·s^−^^1^. The results of phosphoproteomic analysis showed that OCP phosphorylation was accelerated under nitrogen deficiency, suggesting that heat dissipation by OCP is negatively regulated by phosphorylation.

Overall, our results indicate that, in addition to the regulation of protein expression, the regulation of phosphorylation levels of cyanobacterial photosynthetic apparatus and metabolic enzymes is crucial for adapting to changing environmental conditions. In the future, in addition to further detailed analyses of phosphoproteome, elucidation of the phosphorylation control of individual proteins is warranted to clarify the regulatory mechanism of overall photosynthesis, which is expected to allow for the artificial optimization of photosynthesis.

## 4. Materials and Methods

### 4.1. Culture Conditions

*Synechocystis* sp. PCC 6803 GT strain, isolated by Williams [35], was used in this study. The cyanobacteria were grown in modified BG11 medium [36] containing 5 mM NH_4_Cl as a nitrogen source. Cells were grown in 200 mL of medium in 500 mL Erlenmeyer flasks for batch culture under photoautotrophic conditions with continuous light (~50 μmol μm^−^^2^·s^−^^1^) at preculture. *Synechocystis* sp. PCC 6803 (GT) was cultured under the following five conditions (Figure 1a): (1) photoautotrophic (Auto; photosynthesis only), (2) mixotrophic (Mixo; photosynthesis and glucose utilization), (3) heterotrophic (Hetero; glucose utilization only), (4) dark (Dark), and (5) nitrogen-deprived (−N). Cultivation was performed by rotary shaking under five different nutrient conditions at 34°C with continuous illumination by LED lights at 50 μmol μm^−^^2^·s^−^^1^. Under Mixo and Hetero conditions, 5 mM of glucose was added as a carbon source, and under Hetero condition, 10 μM of 3-(3,4-dichlorophenyl)-1,1-dimethylurea (DCMU), a photosynthetic inhibitor, was added. Under −N and Dark conditions, cells were incubated under Auto until confluence (OD ~ 1.0) and then transferred to the respective conditions. Under −N, the cells were collected by centrifugation (4000× *g*, 5 min, room temperature (RT)), and washed with a nitrogen-free medium (BG11_0_) and then re-cultured in the BG11_0_ medium. Samples for proteomic analysis were collected 24 and 48 h after nitrogen deprivation (hereafter referred to as N24 and N48, respectively). Under Dark condition, upon reaching confluence (OD ~ 1.0), the culture was covered with an aluminum foil and then re-cultured in the dark for 6 h. Cells grown under Auto, Hetero, and Mixo conditions were collected when they reached an OD_730_ of ~1.0 (at 96 h, 48 h, and 48 h after inoculation, respectively). Cells grown under Dark condition was collected 6 h after culture in darkness. Cells grown under N24 and N48 were collected 24 and 48 h after being inoculated in BG11_0_ medium.

### 4.2. Sample Preparation for Proteomic Analysis

Total proteins were extracted as described by Picotti et al. [37]. The suspension containing cells in the logarithmic growth phase (200 mL, OD_730_ = 1.0) was collected by centrifugation (5000× *g*, 4 °C, 5 min). Pellets were resuspended in 1 mL of lysis buffer (50 mM HEPES, 15% glycerol, 15 mM dithiothreitol (DTT), 100 mM KCl, 5 mM ethylenediamine-*N,N,N′,N′*-tetraacetic acid disodium salt dihydrate (EDTA), one cOmplete protease inhibitors cocktail per 10 mL (Roche, Mannheim, Germany), and one PhosSTOP (Roche) per 10 mL). The suspension was transferred to an Eppendorf tube containing zirconia beads (diameter, 0.6 and 6 mm) and disrupted with Beads Crusher μT-12 (TAITEC, Saitama, Japan) (3000 min^−1^, 6 min). The resulting solution was centrifuged (15,000× *g*, 4 °C, 5 min), and the supernatant was transferred to a proteomics Eppendorf tube (protein low-adsorption tube) to obtain protein extraction samples. Protein concentration in the extracted sample was measured by the Bradford method, and the total protein content was adjusted to 3 mg. Denatured buffer (500 mM Tris–HCl, 10 mM EDTA, and 7 M guanidine HCl) was added to the adjusted samples to make up a total volume of 2.2 mL.

### 4.3. Reduction and Alkylation/Methanol Chloroform Precipitation

Dithiothreitol (DTT) 1 (50 mg mL^−1^, 10 μL) was added to the protein samples and shaken at RT for 1 h using a tube mixer (CM-1000 Cute Mixer, EYELA, Tokyo, Japan). Next, 25 μL of 50 mg mL^−1^ iodoacetamide was added and shaken for 1 h to reduce/alkylate the proteins. Next, the proteins were purified by methanol/chloroform precipitation, as described by Wessel and Flügge [38]. Cold methanol (6 mL) was added to the sample solution and mixed by inversion. Then, 1.5 mL of cold chloroform was added and mixed by inversion. Cold Milli-Q water (4.5 mL) was added and mixed by inversion, followed by centrifugation (4000× *g*, 4 °C, 5 min). The upper layer was removed, and 450 μL of cold methanol was added and mixed gently by inversion. After centrifugation (4000× *g*, 4 °C, 5 min) using a swing rotor, the supernatant was removed, and an additional cycle of centrifugation (4000× *g*, 4 °C, 1 min) was performed to completely remove the supernatant.

### 4.4. Trypsin/LysC Digestion

Proteins were digested by trypsin and LysC (Promega). Trypsin hydrolyzes the ester bonds on the carboxyl side of Arg, while trypsin and LysC hydrolyze the ester bonds on the carboxyl side of Lys. The combination of trypsin and LysC enhances the efficiency of digestion. Therefore, trypsin/LysC digestion was carried out as described previously [27]. To the supernatant obtained from the above steps, 90 μL of 6 M urea was added and shaken for about 10 min at RT using a tube mixer. Then, 360 μL of 0.1 M Tris-HCl (pH 8.5) was added, and ultrasonic treatment and standing on ice were repeated twice for 30 s using an ultrasonic washer (Branson 2510, Danbury, CT, USA) to resuspend the protein precipitate. Next, 10 μL of 0.5 mg mL^−1^ LysC solution and 25 μL of 1% Protease Max solution (Promega, Nacka, Sweden) were added and mixed by tapping, and the samples were incubated at 25 °C for 3 h. Finally, 10 μL of 0.5 mg mL^−1^ trypsin solution was added and mixed by tapping, and the samples were incubated at 37 °C for 16 h.

### 4.5. Samples Desalination

Sample desalination was performed as described previously [39,40,41]. Milli-Q water (75 μL) and of 50% aqueous formic acid solution (30 μL) were added to the trypsin digestion product, and the mixture was stirred with a vortex mixer and centrifuged (15,000 rpm, 4 °C, 5 min). As a result, 580 μL of the supernatant was obtained. To prepare samples for relative quantitative analysis, ^15^N samples (obtained by culturing *Synechocystis* sp. PCC 6803 in BG11 medium with ^15^NH_4_Cl as the nitrogen source) and ^14^N samples were mixed such that the protein contents were at a ratio of 1:1. The sample volumes were prepared such that the total peptide content could be 300 μg. Tryptic peptide concentrations were measured by the BCA method (Pierce BCA Protein Assay kit, Thermo, Rockford, USA).

The samples were diluted five times with Reagent A (5% acetonitrile, 0.1% formic acid). The diluted samples were desalted by MonoSpinC18 (GL Science, Tokyo, Japan). The column was equilibrated with the same bed volume of Reagent B (80% acetonitrile, 0.1% formic acid) and washed with Reagent A by centrifugation (200× *g*, 10 min, RT). The samples corresponding to 300 μg of proteins was loaded onto the column and centrifuged (200× *g*, 10 min, RT). The column was washed by a bed volume of Reagent A by centrifugation (200× *g*, 10 min, RT) twice. The same volume of Reagent B was added to the samples and centrifuged (200× *g*, 10 min, RT). The prepared samples were dried and solidified using a centrifugal concentrator and frozen until phosphopeptide enrichment.

### 4.6. Phosphopeptide Enrichment

Phosphopeptide enrichment was performed with a TiO_2_ column (GL science) as per the following protocol. First, the TiO_2_ column was equilibrated with 200 μL of Solution A (0.5% trifluoro acid and 80% acetonitrile), followed by 200 μL of Solution B (0.5% trifluoro acid, 80% acetonitrile, and 300 mg mL^−1^ lactic acid), and then centrifuged (100× *g*, 3 min, RT). Second, the desalted samples diluted with 2 mL of Solution B were loaded onto the TiO_2_ column and absorbed on the column by centrifugation (100× *g*, 3 min, RT). The column was washed first with 2 mL of Solution A once and then with Solution B twice by centrifugation (100× *g*, 3 min, RT). Finally, the absorbed phosphopeptides were eluted in 200 μL of 5% ammonium solution, followed by 200 μL of 5% pyrrolidine. The eluted solution was immediately acidified by 400 μL of 5% formic acid (pH, 2~3). The eluted acidified samples were desalted by MonoSpinC18 (GL Science) in the same manner mentioned before. Then, the desalted samples were dried and solidified with a centrifugal concentrator and frozen until nanoLC–MS/MS.

### 4.7. Design of MRM Assays

For quantitative proteomic analysis, 221 proteins related to the central metabolic pathways and photosynthetic apparatus were selected from the Kyoto Encyclopedia of Genes and Genomes (KEGG) database [42]. The amino acid sequences of the target proteins were obtained from Cyanobase [43]. MRM assays used to quantify these 221 proteins designed using the open-source software Skyline 2.6 [44]. Each protein was subjected to a tryptic peptide filter of 8 to 25 residues, and 5 y-fragments (y1 to y5) were selected for each peptide. Samples of *Synechocystis* sp. PCC 6803 were analyzed once by nanoLC–MS/MS (LCMS-8060, Shimadzu, Kyoto, Japan) using the provisional MRM assay. From the results of this analysis, peaks were selected based on the shape, coelution, and intensity, and the best transitions up to 5 were selected. For proteins with no suitable tryptic peptides and transitions, transitions were quantified for all y- and b-fragments, from which the tryptic peptides suitable for quantitation were selected again to design the final MRM assay.

For quantitative phosphoproteomic analysis, the list of targeted phosphoproteins was obtained from previous studies [12,14]. Thus, 101 phosphoproteins mainly involved in photosynthesis, central metabolism, and transcriptional regulation were selected. For designing MRM assays, y- and b-fragments (5 with m/z > precursor) were selected for each phosphopeptide.

Finally, 101 targeted phosphoproteins in *Synechocystis* sp. PCC 6803 were analyzed once by nanoLC–MS/MS (LCMS-8060, Shimadzu) using the provisional MRM assay. From the results of this analysis, peaks were selected based on shape, coelution, and intensity, and the best 5 fragments were selected. The phosphopeptides with fewer than 3 fragments were excluded following analysis. Finally, the MRM assays for 32 phosphoproteins were successfully designed.

### 4.8. NanoLC–MS/MS Using MRM Assays

The obtained phosphopeptides were analyzed on a quadrupole mass spectrometer (LCMS-8060, Shimadzu) as described previously [45]. Electrospray ionization (ESI) was performed, and the sample were separated by nanoLC (LC-20ADnano, Shimadzu). The analytical conditions were as follows: high-performance liquid chromatography column, L-column ODS (pore size: 5 μm, 0.1 × 150 mm; CERI, Tokyo, Japan); trap column, L-column ODS (pore size: 5 μm, 0.3 × 5 mm; CERI); solvent system, water (0.1% formic acid):acetonitrile (0.1% formic acid); gradient program, 10:90, *v/v* at 0 min, 10:90 at 10 min, 40:60 at 45 min, 95:5 at 55 min, and 90:10 at 65 min; and flow rate, 400 nL·min^−1^. MS was performed in the MRM mode, ESI was 1.6 kV, capillary temperature was 150 °C, collision gas was 270 kPa, resolution of Q 1 and Q 3 was low, dwell time was 1.0 ms, pause time was 1.0 ms, and retention time window was 2 min. Each peptide/phosphopeptide was quantified by the peak area ratio of ^14^N samples to ^15^N samples using Skyline 2.6.

### 4.9. Measurement of Ultraviolet–Visible (UV-VIS) Spectra

DU 800 was used to measure the UV-VIS spectra. Absorption spectra of cell suspensions were measured according to the “opal glass method,” with a translucent cuvette placed in front of the detector to minimize the effect of light scattering [46]. The results obtained were normalized to absorbance at 730 nm (1.0).

### 4.10. Proteomic Data Analysis

The results of nanoLC–MS/MS were loaded into Skyline and quantified as the peak ratio of ^14^N samples to ^15^N internal standard samples.

For protein level, each peptide ratio was standardized by the mean of ratios under Auto conditions (*n* = 3). For more than two peptides of a single protein, the mean of all peptides was calculated. For a single peptide of a single protein, the obtained ratio was directly used for quantification. Finally, these parameters were expressed as fold change (FC).

For phosphoprotein level, each phosphopeptide ratio was standardized by the mean of ratios under Auto conditions (*n* = 3). The final FC of each phosphopeptide was calculated as the ratio of FC_phosphopeptide_ to FC_protein_ to subtract the effect of background protein expression. The KEGG database and Cyanobase were referred to assess the function of each protein.

### 4.11. Chlorophyll Fluorescence Measurement

The PAM fluorescence measurements were performed using a fluorescence monitoring system (model FMS1; Hansatech, Norfolk, UK). Aliquots of 2.0 mL of suspension were transferred to the measurement chamber. Cells were maintained in the dark for 15 min. Modulated light was provided with a setting of 1 and an amplifier gain of 70. Actinic light was provided with a power setting of 15 (corresponding to a light intensity of 80 μmol m^−2^·s^−1^) to drive photosynthesis. Pulses of saturating light of 0.2 s duration were applied under power setting of 100 (light intensity 10,000 μmol m^−2^·s^−1^) at 30 s intervals to measure the maximum quantum yield. At the end of each measurement, DCMU (10 μM) was added in the presence of actinic light at a setting 30 to obtain *F*_m_.

## Figures and Tables

**Figure 1 molecules-25-03582-f001:**
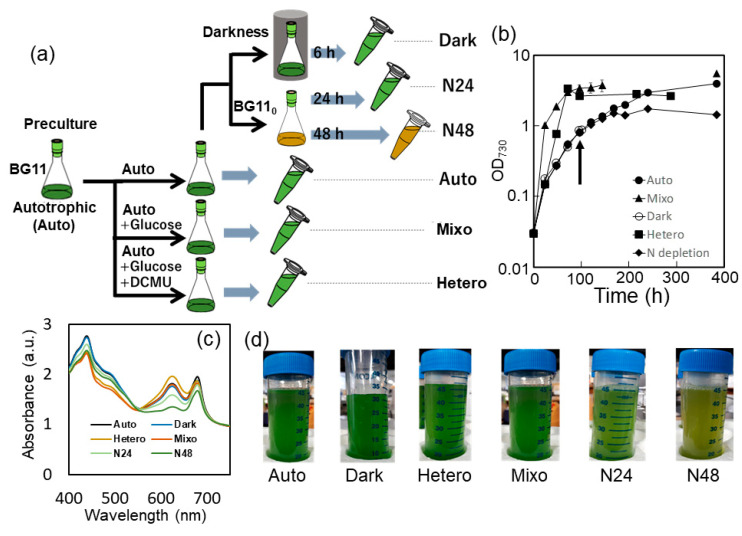
Scheme of cyanobacterial culture and physiological data. (**a**) Workflow of culture under photoautotrophic (Auto), heterotrophic (Hetero), mixotrophic (Mixo), dark (Dark), and nitrogen-deprived (−N: N24 and N48) conditions. BG11 medium contained 5 mM NH_4_Cl as a nitrogen source. BG11_0_ medium is a nitrogen-free medium. DCMU [3-(3,4-dichlorophenyl)-1,1-dimethylureais] is a photosynthetic inhibitor. (**b**) Growth of *Synechocystis* sp. PCC 6803 under Auto, Hetero, Mixo, Dark, and −N conditions. Each value represents the mean ± standard deviation of three independent experiments. Arrows indicate the time the culture was transferred to the dark or nitrogen-deprived conditions. (**c**) Absorption spectra of cell suspensions. The spectra were normalized to absorbance at 750 nm. (**d**) Color of cell suspensions.

**Figure 2 molecules-25-03582-f002:**
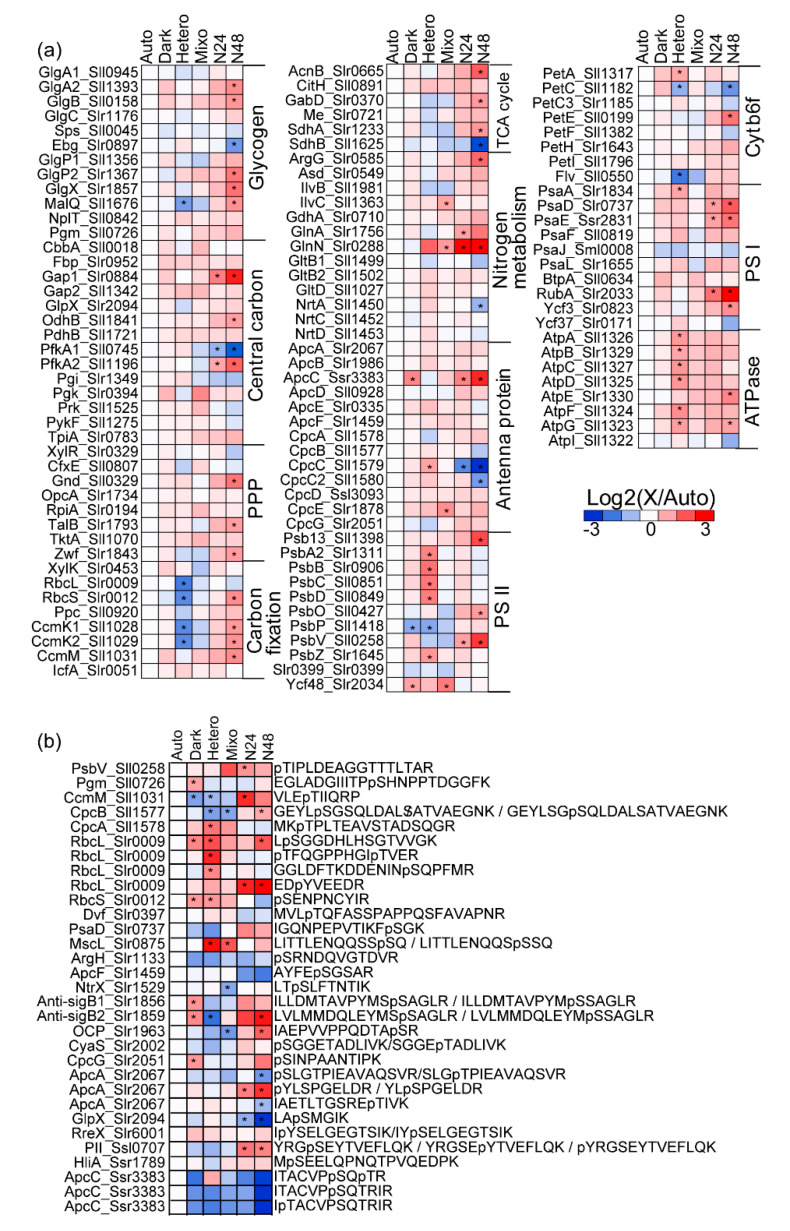
Heat maps comparing protein abundance (**a**) and phosphorylation levels (**b**). Color scale shows the relative abundance of each protein normalized to the value under Auto condition: blue indicates low level, red indicates high level, and white indicates no change. Asterisks (*) indicate a significant change at the level of 5%.

**Figure 3 molecules-25-03582-f003:**
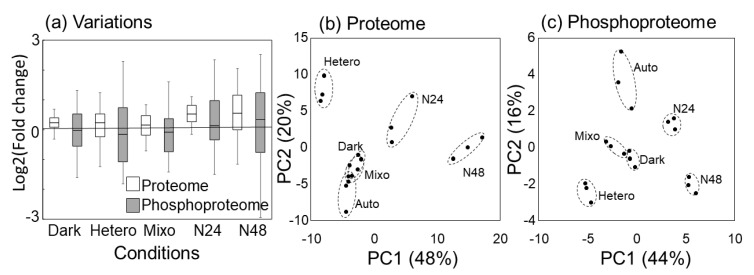
Comparison between the protein expression and phosphorylation profiles. (**a**) Median and interquartile range of the fold change [log_2_(X/Auto)] of protein abundance and phosphorylation level. (**b**) Principal component analysis (PCA) of protein expression profile. (**c**) PCA of phosphorylation level profile.

**Figure 4 molecules-25-03582-f004:**
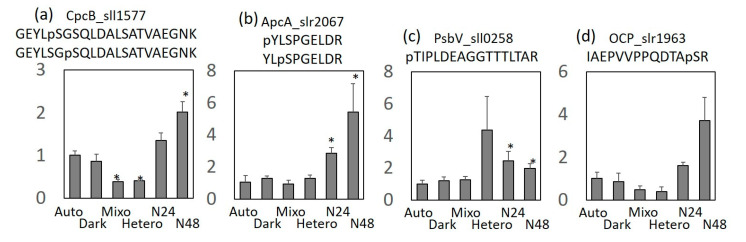
Change in the level of the phosphopeptides. (**a**) GEYLpSGSQLDALSATVAEGNK or GEYLSGpSQLDALSATVAEGNK phosphopeptide of CpcB (**b**) pYLSPGELDR or YLpSPGELDR phosphopeptide of ApcA (**c**) pTIPLDEAGGTTTLTAR phosphopeptide of PsbV (**d**) IAEPVVPPQDTApSR phosphopeptide of OCP. Phosphorylation ratio was normalized at phosphorylation level of Auto. Each value represents the mean ± SD of three independent experiments. Asterisks (*) indicate an increase or decrease at the significance level of 5%.

**Figure 5 molecules-25-03582-f005:**
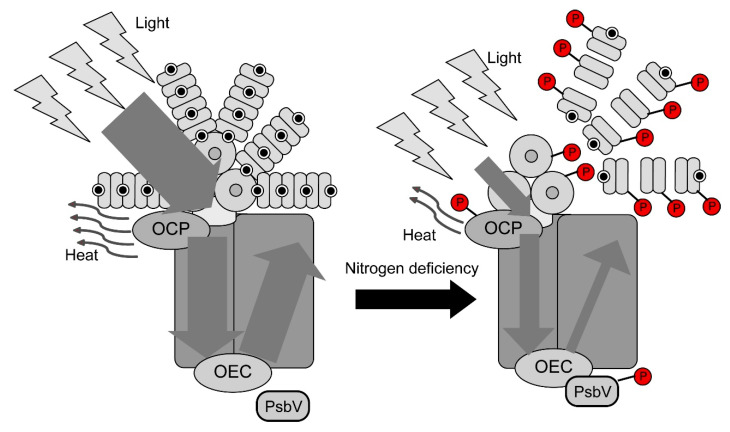
Scheme of phosphorylation and light reaction in *Synechocystis* sp. PCC 6803 under nitrogen deficiency. Under nitrogen deficient conditions, degradation of phycobilisome by CpcB (rod) and ApcA (core) phosphorylation, suppression of PSII photosynthetic activity by PsbV phosphorylation and reduce of heat dissipation by orange carotenoid-binding protein (OCP) occur. And then uptake of light energy and rate of photosynthetic electron transfer are reduced. Abbreviation: OEC, oxygen-evolving complex.

**Table 1 molecules-25-03582-t001:** Photosynthetic performance of *Synechocystis* sp. PCC 6803 under nitrogen deficiency at 24 h and 48 h.

	qP	qN	ΦII	NPQ	*F*_v_/*F*_m_	*F*_v_’/*F*_m_’
Auto	0.194 ± 0.069	0.282 ± 0.032	0.077 ± 0.031	0.065 ± 0.021	0.515 ± 0.012	0.393 ± 0.018
N24	0.085 ± 0.032 *	0.185 ± 0.046 *	0.038 ± 0.015	0.032 ± 0.011 *	0.528 ± 0.019	0.444 ± 0.019 *
N48	0.146 ± 0.055	0.167 ± 0.090 *	0.051 ± 0.020	0.026 ± 0.004 *	0.416 ± 0.063 *	0.356 ± 0.073

Six representative parameters of pulse–amplitude modulated fluorescence analysis are shown. Each value represents the mean ± standard deviation of three independent experiments. qP, an index of photochemical quenching; qN and NPQ, non-photochemical quenching; ΦII, actual quantum yield of PSII electron transport; *F*_v_/*F*_m_, maximum quantum yield of PSII; *F*_v_’/*F*_m_’, PSII maximum efficiency. Asterisks (*) indicate an increase or decrease at the significance level of 5%.

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
