# Peer review of "Assessment of Protein Content and Phosphorylation Level in *Synechocystis* sp. PCC 6803 under Various Growth Conditions Using Quantitative Phosphoproteomic Analysis"

_molecules, 2020, doi:10.3390/molecules25163582_

Round 1
Reviewer 1 Report
In this study, target proteomics was used to quantitatively analyze the phosphorylation levels of specific proteins in Synechocystis sp under various growth conditions. Based on the obtained data, they demonstrated changes in protein expression and phosphorylation levels of the photosynthetic apparatus and metabolic enzymes under these growth conditions. Furthermore, they evaluated the association of regulation of protein expression and phosphorylation with adaptability to different environments. The manuscript is well written, informative and provides new knowledge on phosphorylation levels of proteins in these cyanobacteria under various conditions. I do recommend accepting the paper in the present form.
Author Response
In this study, target proteomics was used to quantitatively analyze the phosphorylation levels of specific proteins in Synechocystis sp under various growth conditions. Based on the obtained data, they demonstrated changes in protein expression and phosphorylation levels of the photosynthetic apparatus and metabolic enzymes under these growth conditions. Furthermore, they evaluated the association of regulation of protein expression and phosphorylation with adaptability to different environments. The manuscript is well written, informative and provides new knowledge on phosphorylation levels of proteins in these cyanobacteria under various conditions. I do recommend accepting the paper in the present form.
A. We would like to express our gratitude to you for your positive comments on this paper.
Reviewer 2 Report
The research article by Toyoshima et al. reveals the regulation of intracellular metabolism and photosynthesis in Synechocystis sp. PCC 6803 under various growth conditions (photoautotrophic, mixotrophic, heterotrophic, dark, and nitrogen-deprived conditions) using targeted and phosphoproteomic approach. The manuscript reads well, and all sections well delineated.
However,
- Line 45-47 – The amino acids most commonly phosphorylated are serine, threonine, tyrosine in eukaryotes but there are also other amino acids that can be phosphorylated. Authors should describe all the possible residues in brief that can be phosphorylated.
- Please define DCMU.
- Line 89…One third should be two third.
- In the Fig 1 A, Hetero should be Mixo and Mixo should be hetero. Please explain BG11 and DCMU in the figure legends.
- In the Fig 2, Please label the left panel as A.
- Please describe the proposed model (Fig 5) in detail.
- Figure S1 legend, Please cite the literature.
Author Response
We would like to express our gratitude to you for your suggestions and comments, which have improved our manuscript. All revised parts are indicated in red color in the manuscript.
1. Line 45-47 – The amino acids most commonly phosphorylated are serine, threonine, tyrosine in eukaryotes but there are also other amino acids that can be phosphorylated. Authors should describe all the possible residues in brief that can be phosphorylated.
A. Thank your suggestion and we mentioned about phosphorylation in prokaryotic cell in Line 45 as follows:” Phosphorylation usually occurs on serine (Ser), threonine (Thr), and tyrosine (Tyr) residues of eukaryotic proteins. In addition, in prokaryotic proteins, phosphorylation also occurs on histidine, arginine, and lysine residues [4].” In Line 66 “In this study, target proteomics was used to quantitatively analyze the phosphorylation levels of Ser, Thr, and Tyr residues of specific proteins.”
2. Please define DCMU.
A. DCMU is an abbreviation for 3-(3,4-dichlorophenyl)-1,1-dimethylurea. We added the official name of DCMU in Line 156.
3. Line 89…One third should be two third.
A. We corrected Line 80 as follows: "two third".
4. In the Fig 1 A, Hetero should be Mixo and Mixo should be hetero. Please explain BG11 and DCMU in the figure legends.
A. We corrected Fig 1a. We swapped Mixo and Hetero correctly. And we added the explanations of BG11 and DCMU in the figure legends as follows: “BG11 medium contains 5 mM NH4Cl as a nitrogen source. BG110 is a nitrogen-free medium. DCMU [3-(3,4-dichlorophenyl)-1,1-dimethylureais] is a photosynthetic inhibitor.”
5. In the Fig 2, Please label the left panel as A.
A. Thank you and we added the label “a” in the left panel of Fig. 2
6. Please describe the proposed model (Fig 5) in detail.
A. We added the explanation of the proposed model in Fig. 5 legend as follows: “Under nitrogen deficient conditions, degradation of phycobilisome by CpcB (rod) and ApcA (core) phosphorylation, suppression of PSII photosynthetic activity by PsbV phosphorylation and reduce of heat dissipation by OCP occur. And then uptake of light energy and rate of photosynthetic electron transfer are reduced”.
7. Figure S1 legend, Please cite the literature.
A. We added the cite “Ref. 18” in the Figure S1 legend.
Reviewer 3 Report
In this work, the author compares the proteomics and phosphoproteomics of Synechocystis in different growth conditions. My concerns are as follows:
1 I didn't see a rational design of this research. Growth was conducted in five conditions, but the author seemed to focus on nitrogen deficiency. An regulation model under nitrogen deprived condition was proposed at the end while other conditions were poorly compared and discussed.
2 What's the sampling time of autotrophic, heterotrophic and mixotrophic cells? Is 6 hours long enough for changes to happen in proteomics level under darkness? A time-course sampling may be more telling?
3 Conclusion in Line 276 is not sufficiently supported by current results.
4 The first paragraph in Results 2.1 should be moved to Materials and Methods.
5 The labels in Fig.2 are too small to read.
6 Switch "Hetero" and "Mixo" in Fig.1 a.
Author Response
We would like to express our gratitude to you for your suggestions and comments, which have improved our manuscript. All revised parts are indicated in red color in the manuscript.
1. I didn't see a rational design of this research. Growth was conducted in five conditions, but the author seemed to focus on nitrogen deficiency. An regulation model under nitrogen deprived condition was proposed at the end while other conditions were poorly compared and discussed.
A. Under heterotrophic, mixotrophic, and dark conditions, small changes in the levels of phosphorylation of individual enzymes were observed. We mentioned about these results of other conditions in second – fourth paragraph in Discussion. However, no systematic control as in the nitrogen deprivation condition was found, and the we mainly focused on the nitrogen starvation condition in the Discussion.
2. What's the sampling time of autotrophic, heterotrophic and mixotrophic cells? Is 6 hours long enough for changes to happen in proteomics level under darkness? A time-course sampling may be more telling?
A. We collected autotrophic, heterotrophic and mixotrophic cells at 96 h, 48 h and 48 h, respectively and mentioned it in line 333. Saha et al. 2016 (Ref. 24) showed mRNA levels of various genes in Synechocystis changed significantly at 1 h after dark. We think that Dark 6 h is reasonable from that point of view.
3. Conclusion in Line 276 is not sufficiently supported by current results.
A. As reviewer pointed out, further analysis is needed to prove this conclusion. So we have proposed a possible mechanism that could be combined with the results of previous studies. We corrected line 270 as follows: “Although analysis of the deficient strains and other factors are required to obtain direct evidence, based on the results of our proteomic and phosphoproteomic analyses as well as PAM measurements, a mechanism for adaptation to nitrogen deficiency was proposed, as illustrated in Figure 5.”
4. The first paragraph in Results 2.1 should be moved to Materials and Methods.
A. Thank you for your suggestion. We deleted the first paragraph in Results 2.1 and corrected the Materials and Methods 4.1 as follows: “Synechocystis sp. PCC 6803 GT strain, isolated by Williams [34], was used in this study. The cyanobacteria were grown in modified BG11 medium [35] containing 5 mM NH4Cl as a nitrogen source. Cells were grown in 200 mL of medium in 500 mL Erlenmeyer flasks for batch culture under photoautotrophic conditions with continuous light (~50 µmol m−2·s−1) at preculture. Synechocystis sp. PCC 6803 (GT) was cultured under the following five conditions (Figure 1a): (1) photoautotrophic (Auto; photosynthesis only), (2) mixotrophic (Mixo; photosynthesis and glucose utilization), (3) heterotrophic (Hetero; glucose utilization only), (4) dark (Dark), and (5) nitrogen-deprived (−N). Cultivation was performed by rotary shaking under five different nutrient conditions at 34°C with continuous illumination by LED lights at 50 µmol m−2·s−1. Under Mixo and Hetero conditions, 5 mM of glucose was added as a carbon source, and under Hetero condition, 10 µM of 3-(3,4-dichlorophenyl)-1,1-dimethylurea (DCMU), a photosynthetic inhibitor, was added. Under −N and Dark conditions, cells were incubated under Auto until confluence (OD ~ 1.0) and then transferred to the respective conditions. Under −N, the cells were collected by centrifugation (4000 × g, 5 min, room temperature [RT]), and washed with a nitrogen-free medium (BG110) and then re-cultured in the BG110 medium. Samples for proteomic analysis were collected 24 and 48 h after nitrogen deprivation (hereafter referred to as N24 and N48, respectively). Under Dark condition, upon reaching confluence (OD ~ 1.0), the culture was covered with an aluminum foil and then re-cultured in the dark for 6 h. Cells grown under Auto, Hetero, and Mixo conditions were collected when they reached an OD730 of ~1.0 (at h, h and h after inoculation, respectively). Cells grown under Dark condition was collected 6 h after culture in darkness. Cells grown under N24 and N48 were collected 24 and 48 h after being inoculated in BG110 medium.”.
5. The labels in Fig.2 are too small to read.
A. Thank you for your suggestion and we corrected Figure 2.
6. Switch "Hetero" and "Mixo" in Fig.1 a.
A. We switched "Hetero" and "Mixo" in Fig.1a.
Round 2
Reviewer 3 Report
I don't have further concerns.